# Effect of Saccharides Coating on Antibacterial Potential and Drug Loading and Releasing Capability of Plasma Treated Polylactic Acid Films

**DOI:** 10.3390/ijms23158821

**Published:** 2022-08-08

**Authors:** Ilkay Karakurt, Kadir Ozaltin, Hana Pištěková, Daniela Vesela, Jonas Michael-Lindhard, Petr Humpolícek, Miran Mozetič, Marian Lehocky

**Affiliations:** 1Centre of Polymer Systems, University Institute, Tomas Bata University in Zlin, Nam. T.G.M. 5555, 76001 Zlin, Czech Republic; 2National Center for Micro- and Nanofabrication, Technical University of Denmark, Building 347 East, Ørsteds Plads, 2800 Kongens Lyngby, Denmark; 3Faculty of Technology, Tomas Bata University in Zlín, Vavreckova 275, 76001 Zlin, Czech Republic; 4Department of Surface Engineering, Jozef Stefan Institute, Jamova Cesta 39, 1000 Ljubljana, Slovenia

**Keywords:** surface functionalization, chitosan, chondroitin sulfate, antibacterial activity, contact killing, biocompatibility, polyelectrolyte complex

## Abstract

More than half of the hospital-associated infections worldwide are related to the adhesion of bacteria cells to biomedical devices and implants. To prevent these infections, it is crucial to modify biomaterial surfaces to develop the antibacterial property. In this study, chitosan (CS) and chondroitin sulfate (ChS) were chosen as antibacterial coating materials on polylactic acid (PLA) surfaces. Plasma-treated PLA surfaces were coated with CS either direct coating method or the carbodiimide coupling method. As a next step for the combined saccharide coating, CS grafted samples were immersed in ChS solution, which resulted in the polyelectrolyte complex (PEC) formation. Also in this experiment, to test the drug loading and releasing efficiency of the thin film coatings, CS grafted samples were immersed into lomefloxacin-containing ChS solution. The successful modifications were confirmed by elemental composition analysis (XPS), surface topography images (SEM), and hydrophilicity change (contact angle measurements). The carbodiimide coupling resulted in higher CS grafting on the PLA surface. The coatings with the PEC formation between CS-ChS showed improved activity against the bacteria strains than the separate coatings. Moreover, these interactions increased the lomefloxacin amount adhered to the film coatings and extended the drug release profile. Finally, the zone of inhibition test confirmed that the CS-ChS coating showed a contact killing mechanism while drug-loaded films have a dual killing mechanism, which includes contact, and release killing.

## 1. Introduction

Despite comprehensive research and development studies, biomaterial devices and implants-related infections are still a major health concern for their long-term use. It is apparent that many microorganisms can attach to a wide range of biomedical instruments and can cause infections, which are also known as nosocomial infections [1]. Apart from the severe prevalence and mortality rates, these infections increase the length of hospital stays and health-care costs. Therefore, actions are needed to be taken to remove or reduce bacteria colonization by tailoring surface properties of biomedical devices that are unfavorable for microorganism attachments. One main reason of biomaterial-associated infection is bio-inert materials that cause bacteria attachment and biofilm formation on the surface. Therefore, one of the methods to for biomaterial-related infections prevention is the usage of immobilized antibacterial agents that can inhibit bacteria adherence or kill them upon contact [1].

For the biomaterials which are susceptible to bacteria colonization, surface modification is an important approach in the fabrication of antibacterial devices and implants. In recent years, biofunctionalization of materials with antibacterial features focuses on the immobilization of bactericidal agents to the biopolymer surface, such as antibiotics, antimicrobial enzymes and peptides, cationic molecules, and metals [2]. However, there are some limitations, including burst release of the antibacterial agents, cytotoxicity issues, and multi-resistance emergence [3].

Strategies to improve antibacterial efficiency of biomaterials can be grouped into two approaches: passive coatings and active coatings [4]. The passive coatings on biopolymers can kill the bacteria upon contact; however, they do not influence the planktonic bacteria dwelling in tissues around the implanted biomaterial. On the other hand, in the case of active coatings the loaded antibacterial agents release to the peripheral tissues by diffusion or dissolution. However, both of these single-function surfaces have their inherent disadvantages [5]. Due to this, various dual-function antibacterial surfaces with both passive and active antibacterial mechanisms together have been developed, showing superior performance for combating bacteria-attached surfaces. Moreover, some infections should not be treated with a single antibiotic because of the risk of triggering a rapid bacteria resistance [6]. A combination of an antibiotic and a natural antimicrobial agent would be an alternative proactive measure in preventing antimicrobial resistance by reducing the usage of high amounts of antibiotics.

Polylactic acid (PLA) is a highly versatile polymer commonly used in biomedical fields for tissue engineering and drug delivery systems, and has exceptionally good advantages over conventional polymers [7]. The US Food and Drug Administration (FDA) have approved PLA products as suitable for direct contact with body liquids. The most important advantages of this polymer are biocompatibility, easy processing, low-cost production, and ecofriendliness [8]. However, one of the major drawbacks of using PLA in biomedical fields is their vulnerability to bacterial invasion [9]. To be able to use as a biomaterial, PLA-based materials should be both biocompatible and antibacterial. The lack of reactive functional groups on the polymer surface renders it chemically inactive. Due to this, before any application on the polymer surface it needs to be activated. Several approaches for activating the PLA surfaces were reported. Among these methods, plasma technology-based treatments as physical surface modification methods are growing in a remarkable way with their numerous advantages [10,11,12].

Plasma post-irradiation grafting is a two-step technique of which the first step of plasma treatment followed by polymerization reactions. While plasma treatment creates temporary, limited lifetime species, post-irradiation grafting method results in a permanent effect [13,14]. In this step, the activated polymer surface is brought into contact with monomers and by immersing into a monomer solution, polymerization will take place by radical polymerization reactions.

The mixing of solutions of polycations and polyanions leads to the spontaneous formation of interpolymer complexes also known as polyelectrolyte complexes (PEC). The driving force for the complexation is Coulomb’s (electrostatic) interactions between oppositely charged microdomains of polyionic components [15]. Other interactions, such as hydrogen bonding, hydrophobic interactions, van der Waals interactions, and dipole interactions, can also contribute to the complexation process [16]. The PEC coatings ameliorate surface properties of the biomaterial due to the interaction between oppositely charged polymers leading to a strong surface modification [17]. Moreover, the PEC properties may differ profoundly from the polyions in their structure [18]. Therefore, such complexes are of great interest as potential drug delivery systems due to the easy integration of charged species into these complex particles [19,20]. PEC coatings have gained increasing interest as ultrathin biologic reservoirs, thanks to the ability readily coat different geometries [21].

The success of biomedical devices, including orthopedic implants and cardiovascular devices, is bound to host tissue response [22]. Therefore, a biocompatible coating of these implanted materials to direct the host cell-surface interface response becomes detrimentally important. Saccharides have been extensively used in biomedical applications thanks to their easy accessibility, biocompatibility, biodegradability, and nontoxicity properties [23]. Polysaccharide-based coatings have been utilized widely to deliver biomedical agents such as antibiotics, antioxidants, proteins, and peptide drugs with the formation of PECs [24]. Chitosan (CS), a cationic polysaccharide, has received great attention in pharmaceutical delivery due to its promising properties, such as it possesses fine biocompatibility, low toxicity, bioresorbability, antibacterial activity, abundant availability, and biodegradability [25,26]. This polysaccharide is primarily composed of glucosamine and N-acetyl glucosamine units bonded with a 1, 4-β-linkage. Besides its exceptional biological characteristics, CS possesses antimicrobial properties [27]. Moreover, in some of the latest studies, it is proposed that the building blocks of CS, which are N-acetyl-D-glucosamine (GlcNAc) and D-glucosamine (GlcN), could strongly re-sensitize drug-resistant bacteria to specific antibiotics by enhancing the susceptibility of tolerant cells [28]. Many polyanions from different types of sources have been broadly investigated to create PEC with CS, including natural and synthetic polymers and metal anions [29]. With the different preparation methods, it is possible to obtain PECs in various forms such as fibrous membranes, hydrogels, beads, films, and micro/nanoparticles [29]. The improvement of the biocompatibility and drug loading ability of CS with PEC formation is somewhat dependent on the polyanionic materials used. Chondroitin sulfate (ChS), an acidic mucopolysaccharide, can form PEC with CS by electrostatic interactions. This anionic saccharide has a wide range of bioactivity including tissue regeneration, cell proliferation, adhesion, antibacterial, and intercellular signaling [30,31,32,33]. Moreover, its low molecular weight, allows it to freely diffuse through high viscosity CS layers, creating an ionic complexation [31].

The combination of natural polysaccharides to fabricate PECs develops an evolving approach for drug delivery and drug-eluting implants. In the present study, the effect of the polysaccharide-based PEC coating on the surface properties of PLA as well as antibacterial activity, biocompatibility, drug-eluting properties, and bacteria killing mechanisms were investigated. For this purpose, following the plasma-treatment activation of PLA surfaces, the molecules of CS were immobilized on the surface through polyacrylic acid brushes (PAA) either electrostatically or carbodiimide coupling. The PEC formation was obtained with the addition of ChS on CS coated PLA through the dipping method. Lomefloxacin was used as a model drug to test the drug loading and releasing properties of PEC coated PLA films. An antibiotic drug was chosen with the purpose of determining the bacteria killing mechanism of the polymeric coating. Besides hydrophilicity and biocompatibility tests, the antibacterial activity of surface modified PLA films was investigated by agar dilution test against *Escherichia coli (E. coli)* and *Staphylococcus aureus (S. aureus)*. Moreover, bacteria killing mechanism was analyzed through zone of inhibition tests.

## 2. Materials and Methods

Low molecular weight CS (50–190 kDa) with 75–85% degree of deacetylation (99%), chondroitin sulfate A sodium salt (60% balance is chondroitin sulfate C), acrylic acid (99%), sodium metabisulfite (99%), and sodium hydroxide (98%) were supplied by Sigma Aldrich (St. Louis, MO, USA). Poly(lactic acid) (PLA) 4032 D in pellets form was supplied from Nature Works (Blair, NE, USA). N-(3-Dimethylaminopropyl)-N′-ethylcarbodiimide hydrochloride (EDC) 98%, N-Hydroxysuccinimide (NHS) 98% and lomefloxacin were purchased from Sigma Aldrich (St. Louis, MO, USA). All reactants were of analytical grade, and used without further purification.

### 2.1. Preparation of PLA Films and Surface Functionalization

PLA films were prepared as previously described with the compression molding technique [34]. Briefly, dried PLA pellets were hot-pressed at 180 °C with a pressure of 100 kN for 20 min and then cooled. The square-shaped sheets were cut into pieces of 25 × 25 mm for further surface treatments and were rinsed with a detergent solution. Then, dried PLA films were exposed to radio frequency (13.56 MHz) low-pressure plasma treatment (PICO Diener, Ebhausen, Germany) at reactor power of 50 W for 60 s, under 60 Pa vacuum chamber and 20 sccm of airflow.

Immediately after plasma treatment, PLA films were immersed into sodium metabisulfite containing 10% acrylic acid (AAc) solution for 24 h for obtaining polymeric brushes on the PLA surface with the purpose of enhancing the reactivity characteristic of the PLA surface with the bioactive material that was attached on it. Polymeric brushes have been used in many biological applications thanks to their well-defined architecture, highly accessible functional groups, and the ability to immobilize a variety of substances [35]. For hydrophilic brushes on the PLA surface, the radical polymerization induced ‘grafting from’ method was applied. Radicals and peroxides on the PLA surface initiate graft polymerization of PAA [36,37,38].

PLA films, having PAA brushed through the ‘grafting from’ method, were then immersed into 1 w% sodium hydroxide for neutralization. For covalent grafting of CS on the PLA surface, samples were immersed into EDC/NHS solution for 4 h for the activation of PAA carboxyl groups [39]. These coupling agents are also known as zero-length crosslinkers and are mainly used in crosslinking of amines and carboxylates by promoting covalent bond formation [40]. It is called zero-length crosslinkers because after ester formation between these two functional groups, carbodiimide agents leave the reaction as by-products of urea derivatives [41,42]. Firstly, EDC reacts with the carboxyl groups of the PAA to form an active o-acylisurea intermediate product. This intermediate is unstable in aqueous solutions and to increase its lifetime and efficiency, EDC couples NHS to a carboxyl group and forms an amine reactive NHS-ester. Finally, this intermediate ester structure is easily displaced by nucleophilic attack from the primary amino groups of CS in the aqueous solution. This amino group created an amide bond with the original carboxyl group, and the reaction by-product is released as a urea derivative [16].

Thereafter, saccharide assembly was performed as depicted in Figure 1. For the drug release experiments, CS coated samples were dipped into Lomefloxacin-containing ChS solutions. Subsequently, coated/loaded films were then washed 2–3 times with deionized water and dried overnight at 25 °C for further characterization.

### 2.2. Characterizations

X-ray photoelectron spectroscopy (XPS) was used to the evaluate elemental composition of surface-modified PLA films. Thermo Scientific (Waltham, MA, USA) K-Alpha XPS system equipped with a monochromic Al Kα X-ray source (1486.6 eV) was operated at a power of 72 W with a spot size of 400 μm. Data were analyzed using the software Avantage (Thermo Scientific Waltham, MA, USA). The graphs were plotted using OriginPro 8.5.

The surface modification induced wettability change in untreated and treated films was measured by static contact angle measurements (SEE System; Advex Instruments, Brno, Czech Republic). The system was equipped with a charged-coupled device (CCD) camera system to precisely capture the digital images of the drop size of the used liquid. Deionized water was used as the testing liquid and the measurements were conducted at 20 °C and 50% relative humidity. A 5 μL droplet from a micropipette was gently settled on six different spots of the sample surface and the captured images were analyzed by the SEE System software for the contact angle values.

The morphology of the films after plasma treatment and coatings was examined by NANOSEM 450 (FEI, Hillsboro, OR, USA) scanning electron microscope, operating at 5.0 kV accelerating voltage. Before SEM observation, the films were gold/palladium metalized with a thickness of ~20 nm by a sputter coater SC 7620 (Quorum Technologies, Newhaven, East Sussex, UK).

### 2.3. Biocompatibility Testing

The cytocompatibility property of the surface functionalized PLA films was analyzed through a colorimetric assay. A purple formazan reaction between 3-[4,5-Dimethylthiazol-2-yl]-2,5-diphenyltetrasodium bromide (MTT) and the mitochondria of mouse embryonic fibroblast cells (NIH/3T3, ATCC^®^ CRL-1658TM, Merck, Darmstadt, Germany) was used to estimate cell viability.

The ATCC-formulated Dulbecco’s Modified Eagle’s Medium (Biosera, France) supplemented with calf serum and antibiotic (PAA Laboratories GmbH, Pasching Austria) was used as the culture medium. The cells were seeded on UV-sterilized PLA film samples with the density of 1 × 10^5^ cells/mL and the culture medium was added for incubation at 37 ± 1 °C for 72 h. Then, the supernatant of each well was replaced by a fresh medium which included 100 µL tetrazolium dye MTT solution (5 mg mL^−1^). For the dissolution of the formed formazan crystals on the sample surfaces, dimethyl sulfoxide (Merck, Darmstadt, Germany) was added, and the absorbance was recorded on a microplate reader (Infinite M200 Pro NanoQuant, Tecan, Switzerland) using a test wavelength of 570 nm and a reference wavelength of 690 nm.

### 2.4. Antibacterial Activity

The antibacterial performance of PLA samples was quantitatively evaluated against two different bacteria strains, Gram-positive bacteria *Staphylococcus aureus* (*S. aureus*) and Gram-negative bacteria *Escherichia coli* (*E. coli*). Briefly, the number of test bacteria in nutrient broth (NB) was adjusted to certain concentrations, which is between 2.5 × 10^5^ cells/mL and 1.2 × 10^6^ cells/mL, for use as test inoculations. Both pristine and functionalized PLA samples (25 mm × 25 mm) in Petri dishes were completely covered by 0.4 mL of bacteria suspension and incubated for 24 h at 35 °C. After the incubation, the samples were washed with 10 mL of neutralizing broth and the recovered, diluted bacterial suspensions were plated onto 15 mL agar-filled Petri dishes using the spread plate method. Finally, following the 24 h incubation, the number of bacteria colonies were counted.
R = U_t_ − A_t_
(1)
where U_t_ is the average of the common logarithm of the number of viable bacteria, in cells/cm^2^, recovered from the untreated test specimens after 24 h, A_t_ is the average of the common logarithm of the number of viable bacteria, in cells/cm^2^, recovered from the treated test specimens after 24 h.

### 2.5. Inhibition Zone Assay

For the inhibition zone assay, the disk diffusion method was used according to EUCAST (Kirby–Bauer method). The film samples (10 × 10 mm), two different sample squares in one Petri dish, were placed on *E. coli* or *S. aureus* Mueller–Hinton agar plate at an inoculum concentration of 10^8^ colony-forming units per mL (cfu/mL), and then the plates were incubated at 35 °C for 18 to 24 h. After the incubation, the width of the zone inhibition for each sample was measured to the nearest millimeter. The test with each sample was performed in triplicate. Any releasing of the antimicrobial agents would result in an inhibition zone without bacteria growth around the PLA film samples.

### 2.6. In Vitro Drug Release Studies

Drug entrapment properties of the saccharide-grafted PLA films were evaluated by a Photolab 6600 UV-VIS photometer (Xylem Analytics Germany Sales GmbH & Co. KG, Weilheim, Germany) by calculating the amount of lomefloxacin present per unit cm of PLA films. Briefly, for drug loading efficiency, CS-D and CS/ChS-D samples with an area of 2 cm^2^ were placed into 10 mL of acetic acid solution (1% *v*/*v*) and stirred for 24 h on an oscillating stirrer. The absorbance of the withdrawn solutions (1 mL) was analyzed at 285 nm and the lomefloxacin contents of the films were calculated through the calibration curve (Abs = 0.0814 (μg mL^−1^); R^2^ = 0.9991).

The in vitro drug release properties of the films were determined in PBS-filled (pH 7.4) glass vessels at 37 °C to imitate the in vivo environment. The aliquot samples (1 mL) were withdrawn from the PBS solutions at the predetermined time intervals and the amounts of released lomefloxacin were spectrophotometrically determined at a wavelength of 285 nm.

## 3. Results and Discussion

### 3.1. Surface Morphology and Characterization of PLA Films

SEM measurements were carried out to confirm the surface functionalization of the PLA film surfaces. The SEM image of the neat PLA film (Figure 2a) clearly shows that the surface was homogenous and smooth after the compression molding process. It could be seen in Figure 2b that the RF plasma treatment caused rough and micro-crack formations on the PLA surface. That behavior is attributed to the oxidation and etching/ablation properties of the topmost layers of polymers during low-pressure plasma treatment and has been observed in other studies as well [43,44].

In Figure 2c,d, CS and CS-ChS coated PLA film surfaces can be seen, respectively. Chemical grafting of CS through EDC/NHS coupling onto post-plasma treated polymer surface resulted in a thin layer of CS deposition on the surface and the CS immobilization subsequently introduces independent globular structures. A similar observation of morphological arrangements has been stated for the CS immobilization on polyacrylic acid grafted PET surfaces [45].

On the other hand, PEC formation between CS and ChS can be seen as flocculated and coalesced particles on the polymer surface. These relatively non-uniform flocculated clusters are the result of the interaction of a weak polycation and a strong polyanion [46]. Similar occurrences on coating surfaces resulting from polyelectrolyte complex formation have been reported previously. Safitri et al. studied the PEC formation between pectin and CS and reported that upon the complex formation the surface was found to be covered with small granules [24]. Moreover, Cui et al. reported that the film consisted of polyelectrolyte complexes of PAA and polyethylene imine displayed layers of protrusions and porous structures [47].

The effect of surface modification on hydrophilicity was analyzed through static contact angle measurements (Table 1). The average of the static water contact angle for pristine PLA samples was 81.5°, which is in accordance with the literature [48]. When the samples were subjected to plasma treatment, the contact angle reached 46.1° resulting in hydrophilic PLA surfaces. This reduction can be attributed to the surface etching and formation of hydrophilic oxidative functional groups on the polymer surface [49,50]. Upon CS polysaccharide adhesion, the mean contact angles of samples slightly increased due to a decline in the oxidative functional group content. When compared to electrostatically grafted surfaces, carbodiimide coupled CS samples had a higher contact angle value, which is attributed to the increased amount of CS polymer chains grafted on the surface. On the other hand, ChS presence significantly reduced the contact angle to 50.9°. This decrease was expected as ChS is a more strongly charged molecule than CS [51], which increased the charge density on the surface that can result in increased hydrophilicity. Together with SEM images, these preliminary results indicate the successful modification of the PLA surfaces.

For further confirmation of the surface modification, XPS measurements were carried out before and after the treatments. The two sharp peaks are found at 284 eV and 531 eV in the spectrum of the pristine PLA that correspond to C1s and O1s (Figure 3a). After surface modifications, a new peak appears at around 400 eV, which corresponds to N1s, proving the effectiveness of the amine group on the surface both due to the air plasma treatment and saccharide coating on the surface. It was established that sulfur peaks appear only on the samples coated with ChS and fluor only on the drug-loaded samples. In addition, some minor impurities of calcium on untreated, plasma treated, and CS coated PLA films could be due to the rinsing steps.

In order to evaluate the changes in functional side groups before and after modifications, the C1s and N1s spectra were deconvoluted with Avantage software as shown in Figure 4. The C1s spectrum of the untreated PLA surface contained three peak components and the peaks at 283.73 eV, 285.83 eV, and 287.79 eV are attributed to C-C/H, C-O, and O-C=OH/R bond, respectively. A significant increase in the C-O and COO species revealed that atmospheric plasma treatment in air resulted in an almost two-fold increase in oxygen-rich functional groups on the PLA surface. These increases in the molar concentration of C-O and COO species suggested the incorporation of hydroxyl, peroxyl, ether groups, or carbonyl groups on the polymer surface. These results are in agreement with the literature [52,53]. Meanwhile, a new separated peak at 288.18 eV was observed after CS grafting onto polymer surface, which was attributed to -O-C-O- and -N-C=O groups of the introduced polysaccharide. Moreover, CS incorporation reduced the peak intensity of C=O bond as compared with plasma treated films, due to the coupling reaction of CS onto the PLA films. For ChS incorporated films (*g-*CS-ChS), while the peak intensities of COO species significantly reduced, C-O, -O-C-O-, and -N-C=O group-related peak intensities increased.

Figure 5a–c show the N1s core-level spectra, and two peaks occurred in Figure 5a indicating that the plasma treated sample contained two nitrogen regions with binding energies of 399.46 and 398.28 eV which corresponded to N-H and C-N, respectively due to the nitrogen content of the plasma processing gas (air) [54]. With CS coating, a new peak appeared at 402.18 eV, which was attributed to the protonated amine of the polysaccharide. Compared with CS coating, the protonated amine decreased whereas the amide bond slightly increased in CS-ChS coating because of the electrostatic interaction between the two saccharides and amide groups in the ChS structure.

As shown in Table 2, the atomic composition of PLA significantly changed after plasma treatment (RF), resulting in the increased amount of oxygen species and presence of nitrogen content that have the effect of decreasing the relative carbon content due to oxidation and etching of the surface. During the plasma treatment, the oxygen and nitrogen in the carrier gas become short- and long-lived reactive oxygen (ROS) and nitrogen (RNS) species that react with molecular chains of PLA and introduce functional groups to the polymer surface [55,56]. XPS results revealed that sulfur (S) and fluor (F) elements were only detected in ChS coated and drug loaded samples, which prove their presence on the PLA surface.

The grafting protocols of CS were performed with and without EDC/NHS coupling. In the case of carbodiimide coupling (*g*-CS), the amount of nitrogen increased to 3.6%, along with an increase in the N/C ratio. These changes suggest that the coupling made it possible to graft a higher amount of CS to the PLA surface. The ratios of N/C and the amount of nitrogen on the *g*-CS-ChS samples reach 0.164 and 10.1%, respectively, which are significantly higher than those of CS, indicating that ChS has been grafted with CS to form a PEC complex with higher nitrogen content. Moreover, sulfur was found on the assembled polymer films, which was another proof of successful ChS grafting. The presence of lomefloxacin on the polysaccharide-coated film surface was confirmed with the appearance of fluor that is only found in this fluoroquinolone antibiotic drug.

### 3.2. Cytotoxicity of the Films

Figure 6 shows the biocompatibility of pristine and modified polymer films through the direct contact study on NIH/3T3 cells. As can be seen, all modified PLA samples have higher biocompatibility than the neat films. Moreover, RF plasma treatment enhanced the biocompatibility of PLA film due to the increased hydrophilicity of the surface, which was consistent with the literature [57]. It is stated that surface hydrophobicity of a material causes lower cell viability [58,59,60,61]. When a material is placed in a serum-containing cell medium, protein adsorption occurs very rapidly, at the millisecond level, and by the time cells arrive on the surface, a protein layer has already been formed. However, if the conformation of the proteins has changed, this signal is recognized by cells as a foreign object and causes foreign body reactions. This conformation change occurs mostly on the hydrophobic surfaces because proteins expose their inner hydrophobic parts to adhere onto those surfaces [61,62,63]. Hydrophobic surfaces generally cause more protein adsorption than the hydrophilic ones, because displacement with interfacial water molecules is energetically less favorable on a hydrophilic surface [64,65,66]. Therefore, through the state of the adsorbed proteins, cells can obtain surface information translated by the protein layers and take appropriate countermeasures such as removing, engulfing, or being compatible with the material [67,68].

On the other hand, slightly lower cell viability was observed for both types of CS grafted samples when compared to the plasma-treated ones. This slight reduction in viability may be related to the remnants of acrylic acid monomers. Nevertheless, the % cell viabilities of CS-coated samples were higher than the untreated PLA films. Another remarkable increase in biocompatibility was seen with the carbodiimide coupling of CS with PAA brushes. This effect is probably caused by the higher amount of immobilized CS chains on the surface, which is consistent with the XPS results. Moreover, EDC/NHS coupling reaction has no potentially cytotoxic residuals and does not incorporate into the crosslinked structure. This is because the carbodiimide conjugation activates the carboxyl groups on PAA brushes for direct reaction with the primary amines of the CS backbone and this type of bonding is called zero-length crosslinking [69]. As a result of this reaction, positive charge of the polymer decreases which leads to less binding affinity with the negatively charged cell membranes [70,71,72,73,74].

The highest biocompatibility was observed with the samples of ChS immobilization on CS-grafted polymer surface by electrostatic adsorption. This phenomenon could be ascribed to the masking effect of positive charge on CS with the negatively charged sulfate groups of ChS [31,75]. Cytotoxicity test results were consistent with the hydrophilicity degree of the samples. The most hydrophilic samples had the highest % cell viability. As reported in the literature, while proteins can maintain their conformations on hydrophilic surfaces, hydrophobicity causes protein denaturation [76]. As a whole, the obtained results in this study, indicate that ChS- and CS-coated PLA films do not cause toxicity in the cells and are highly suitable materials for biomedical applications. Moreover, in the literature, it was stated that the lomefloxacin is slightly cytotoxic to cells [77,78]. As can be seen in the drug-loaded CS-ChS coated films, even though the cell viability dropped by almost 40%, due to the high biocompatibility feature of these saccharides’ combination biocompatibility was observed at around 100%. The data implied that this combination has the potential to reduce the cytotoxicity of certain drugs.

### 3.3. Antibacterial Efficiency of PLA Films

The antibacterial efficacy of the neat PLA and modified films was evaluated against Staphylococcus aureus and Escherichia coli strains and analyzed by calculating the antibacterial activity value (R). According to JIS Z 2801 standard, the samples are considered to have antibacterial activity with the value of R equal or superior to two. As can be seen in Figure 7, untreated PLA did not possess antibacterial activity against both the bacteria strains. The bactericidal activity of PEC-coated PLA films (*g*-CS-ChS) against both the Gram-positive and the Gram-negative bacteria strains was higher than the CS-coated ones. The antibacterial property of ChS has been reported in previous studies [79,80]. The improved antibacterial activity with CS-ChS PEC compared to CS alone may be due to an additive effect of both ChS and CS interaction. In our previous study, it has been shown that ChS possesses significant bacteria killing property for both the Gram-positive and negative bacteria and is more effective on *E. coli* than *S. aureus* [34]. In the present study, while CS coated samples expressed more than 1.5-fold higher antibacterial activity on *S. aureus* than *E. coli*, PEC-coated PLA films showed almost equally high bacteria killing property against the both of the bacteria strains and increased overall efficiency, which shows the contribution of ChS coating. The exact antibacterial mechanisms of these polysaccharides are still unclear and for explaining the mechanism of antibacterial activity of CS, different hypothesis have been proposed in the literature. It has been reported by other authors that CS is unable to penetrate the bacterial cell wall and shows its bacteria killing property by acting as a chelating agent and selectively binding to the trace metals that are crucial for stability of the bacteria cell wall [81,82]. Thereby a possible mechanism for ChS enhancing the activity of CS may be by a cooperative interaction between both compounds, which act together on the bacterial membrane by possessing opposite charges and effecting the membrane integrity. One of the explanations for the possible mechanism of antibacterial activity of sulfated polysaccharides, such as ChS, fucoidan, and dextran, is trapping nutrients, for example, cationic minerals, by negatively charged groups in their structure, and resulting in a decrease in the bioavailability of nutrients for microorganisms [83,84,85]. In another study, Sharma et al. investigated the antibacterial properties of ChS and CS scaffolds and found that when they were combined, the formed PEC showed an additive killing effect on the bacteria species [31]. However, further studies are needed to clarify the action mechanisms of these polysaccharides against microbial growth.

CS and ChS have differing inhibitory efficiency against Gram-positive and Gram-negative bacteria because of the differences in cell wall and cell membrane chemistry. In previous studies with CS, greater antibacterial activity was more evident against Gram-positive bacteria than Gram-negative [86,87]. It is stated that in the presence of CS cytoplasm, *E. coli*, which is a Gram-negative bacteria that consists of an outer membrane and a thin peptidoglycan layer, was concentrated, while the cell membrane of *S. aureus,* which is a Gram-positive bacteria that possesses a thick peptidoglycan layer, was weakened and even lost its integrity [87]. The higher susceptibility of Gram-positive bacteria to CS can be explained by the greater negative charge on the cell surface due to the teichoic acid backbone, which can establish an electrostatic interaction with cationic antibacterial compounds [88]. Also in the literature, it is stated that Gram-negative bacteria could be more sensitive to sulfated polysaccharides [89]. The reaction seems to alter the bacterial cell membrane that affects the selective permeability and inhibits the membrane synthesis [85]. In addition, the CS samples grafted through carbodiimide chemistry (*g*-CS) showed better bacteria-killing properties than the ones directly immobilized on PAA brushes (CS). This may be due to the higher amount of CS chains on the PLA surface. With the EDC/NHS reaction mechanism, carboxyl groups on PAA brushes activated and formed a robust amide linkage with the amino groups on CS chains [90]. Although plenty of studies have reported the materials grafted with PAA brushes to achieve an enhanced immobilization of cationic molecules, few works paid attention to the effectiveness of carbodiimide reaction-assisted grafting on antibacterial efficiency and biocompatibility. Finally, with the antibiotic-loaded samples, antibacterial efficiency reached the highest level. In summary, our results demonstrate that CS-ChS combination reduced the antibiotic amount required for bacterial killing with showing 99.99% bacteria destruction.

### 3.4. Inhibition Zone Assay

Possible bacteria killing mechanism of modified PLA films was examined by the zone of inhibition test. As shown in Figure 8, the samples without drug loading were lack of inhibition zones and only lomefloxacin-loaded samples (no.5) showed the bacterial inhibition zone for both of the bacteria strains. The inhibition zones for *S. aureus* and *E. coli* were 5.33 ± 0.94 and 6.67 ± 1.25 mm, respectively. The different zone of inhibitions of the samples no.5 could be due to the inhomogeneous drug loading during complex formation between CS and ChS. In terms of the surrounding clear zone around drug-loaded CS-ChS coated samples, our results have revealed that these samples show a dual mechanism with contact killing through CS-ChS molecules and release killing by leaching of lomefloxacin from the CS-ChS coated surface. It is stated in the literature that CS does not diffuse on the agar gel [91], and our study supports this statement as no inhibition was found beyond the limits of CS-coated PLA films. These experiments also showed that the zone of inhibition tests alone may not be enough to prove antibacterial efficiency of the materials as the agar dilution methods indicated bactericidal activity for CS and CS-ChS coated films, in the zone of inhibition tests antibacterial activity was not observed due to the lack of a clear zone around the samples. Only drug-loaded films showed a clear inhibition zone around the specimens, which indicates a diffusion-related killing mechanism. An antibiotic drug was used as a model drug to demonstrate the contact killing mechanism of polysaccharides coating.

### 3.5. Drug Release

To test the drug loading and release properties of the coating, lomefloxacin is used as a model drug and the results are presented in Figure 9. It can be seen, that the amount of lomefloxacin loading is highest for the PEC-coated PLA films (*g*-CS-ChS). The amount of drug adhering to the CS-coated samples doubled after CS-ChS coating. The results were in line with the literature that shows the PEC formation between polysaccharides increases the drug loading efficiency [92]. The drug loading study showed that direct loading onto plasma treated PLA samples resulted in insufficient drug amount on the surface. On the other hand, polysaccharides coating facilitated higher drug adherence.

The polyelectrolyte complex formation between CS and ChS affects the cumulative release of the drug from the films (Figure 9). The CS-coated films reached a 69.2% drug release rate in 4 h, while the PECs-formed films released only 32.7% of the loaded drug. The lower release rate was observed by the PECs-coated films as expected. The interaction between CS chains and ChS molecules caused entrapment of the drug molecules between complex structures [93].

Moreover, about 16% of lomefloxacin released from the CS-ChS coated films during the first hour, significantly less than those of CS-coated (34.7%) ones. The burst release for both types of films at the beginning of the release time is due to the presence of weakly adsorbed drug molecules on the film surfaces. However, PEC coating lowered the burst release of the drug molecules. An ideal antibacterial coating for implants based on the release of an antibiotic should start releasing at the time of implication and followed by prolonged release over the time for at least some hours [30]. The presented release results showed that the PEC structure directly affects the release profile of lomefloxacin. This implies that the electrolyte complex structure between CS and ChS showed a prolonged release profile.

## 4. Conclusions

In the present study, surface modification of PLA films was achieved through polysaccharides coating following the plasma treatment. Individual and combined effects of both saccharides on antibacterial properties, biocompatibility and drug loading/releasing profiles were evaluated. Together with SEM images, XPS results and contact angle values verified the surface modification of the PLA films. The direct immobilization of CS on PAA brushes was compared with the EDC/NHS assisted coating. From the XPS results, it was clear that with the carbodiimide reaction between PAA and CS chains, the amount of immobilized CS molecules increased significantly. This increase resulted in better antibacterial activity and biocompatibility for the films grafted through carbodiimide chemistry between the brushes and CS molecules. Moreover, the films coated with CS-ChS exhibited better bactericidal properties and higher cell viability than the CS-coated film. The zone of inhibition assay proves that the antibacterial mechanism for CS-ChS coating was contact killing-based and the antibiotic-loaded films showed the dual killing mechanism together with release killing. It was clear that the combined coating of both polysaccharides resulted in significantly higher bioactivity. Besides, from the results of the drug loading study, it could be concluded that the drug amount was mainly dominated by the CS-ChS interaction. The release profile showed that PEC formation between these two polysaccharides resulted in a prolonged drug-release rate. Finally, it is proved that CS and CS-ChS coatings on PLA films possess contact killing ability while drug-loaded films showed zones of inhibition, which indicates a dual killing mechanism that covers both release and contact killing routes.

## Figures and Tables

**Figure 1 ijms-23-08821-f001:**
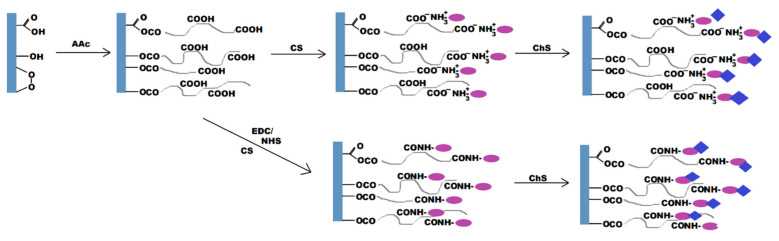
Plasma post-irradiation grafting of acrylic acid (AAc) onto the polylactic acid (PLA) surface followed by immobilization of chitosan (CS) through either direct or carbodiimide coupling and chondroitin sulfate (ChS) molecules immobilization.

**Figure 2 ijms-23-08821-f002:**
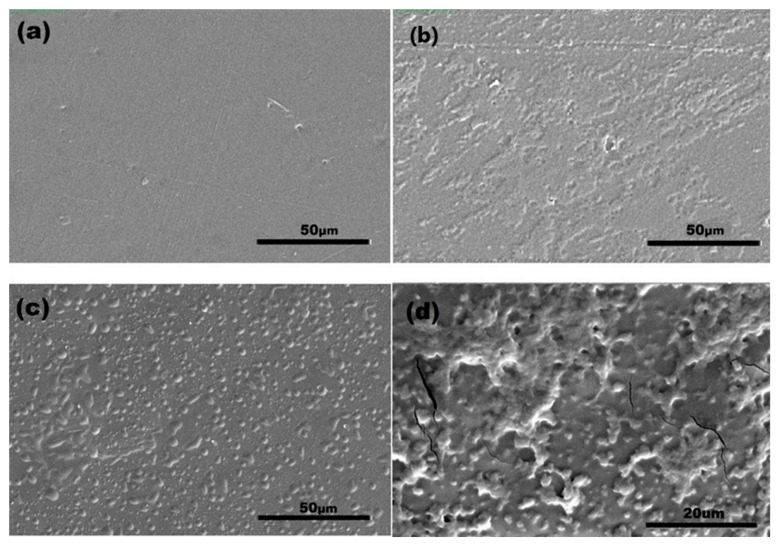
SEM images of the (**a**) untreated PLA; (**b**) plasma treated PLA; (**c**) CS; (**d**) CS-ChS grafted films.

**Figure 3 ijms-23-08821-f003:**
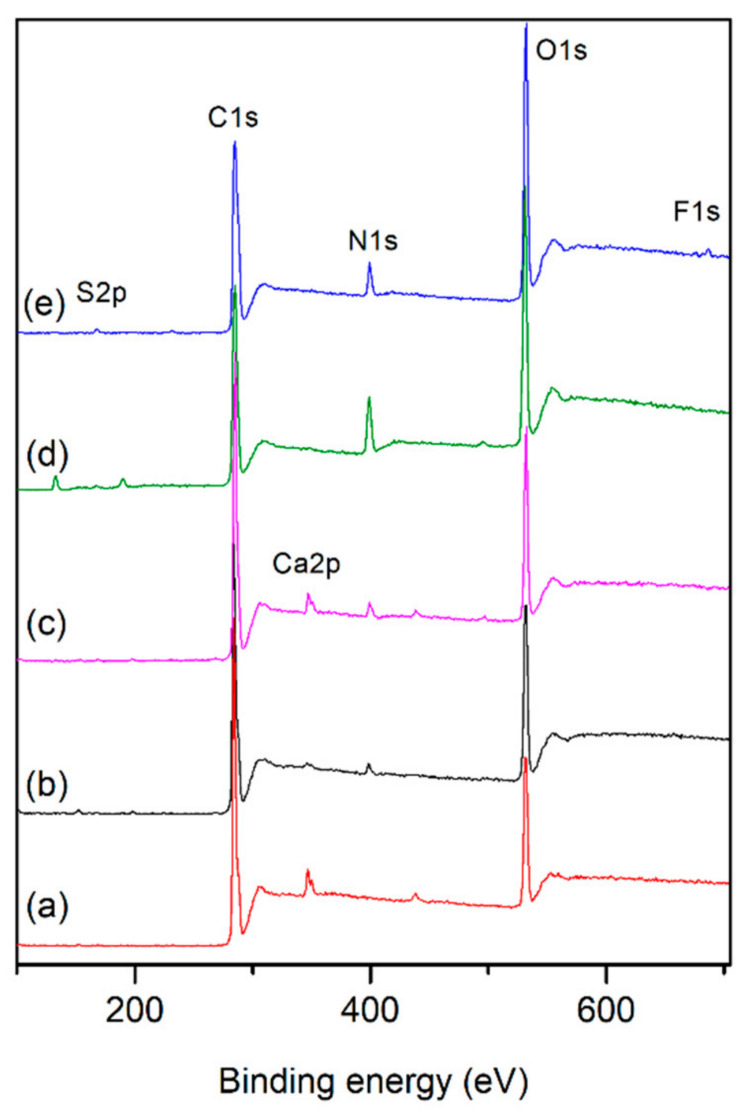
XPS spectra of (**a**) untreated PLA, (**b**) CS, (**c**) *g*-CS, (**d**) *g*-CS-ChS, (**e**) *g*-CS-ChS-L.

**Figure 4 ijms-23-08821-f004:**
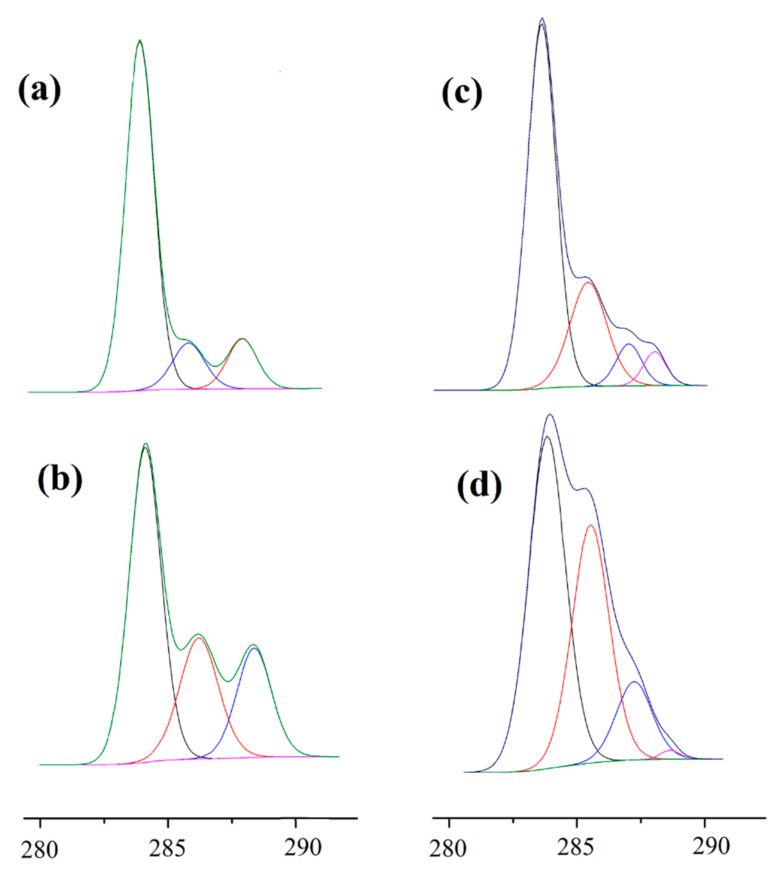
XPS C1s spectra of (**a**) untreated PLA, (**b**) RF, (**c**) *g*-CS, (**d**) *g*-CS-ChS.

**Figure 5 ijms-23-08821-f005:**
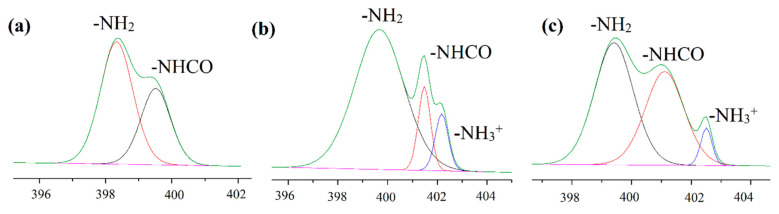
XPS N1s spectra of (**a**) RF, (**b**) *g*-CS, (**c**) *g*-CS-ChS. The green line corresponds to the envelope/fitting curve, and the colored curves to the deconvoluted components.

**Figure 6 ijms-23-08821-f006:**
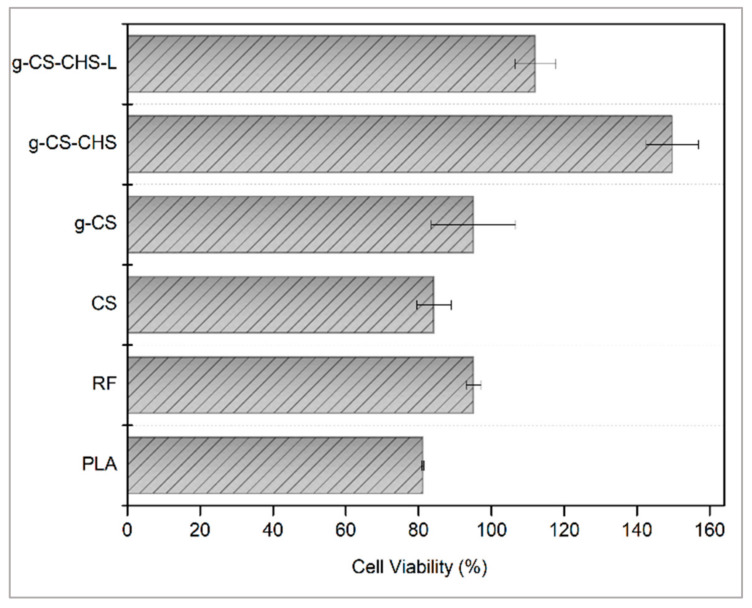
Cytotoxicity of untreated, plasma treated, CS and/or ChS coated, lomefloxacin loaded PLA samples.

**Figure 7 ijms-23-08821-f007:**
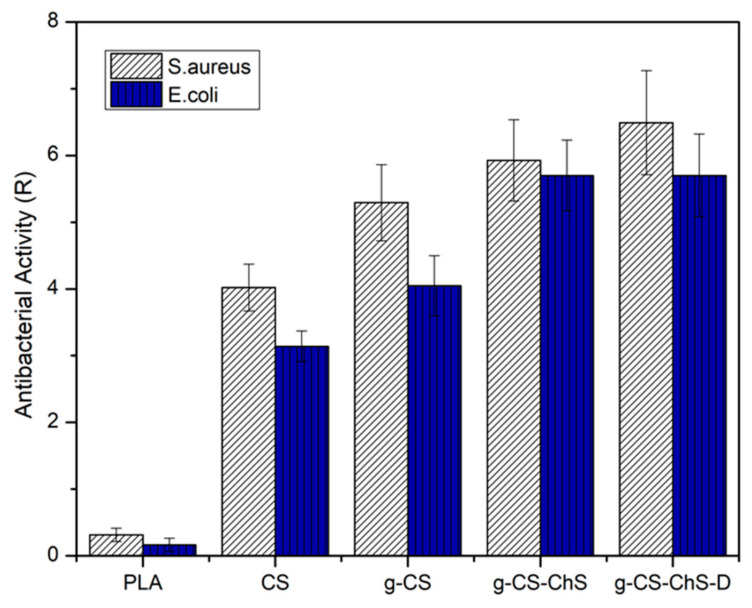
Antibacterial activity of untreated, CS and/or ChS coated, lomefloxacin loaded PLA samples.

**Figure 8 ijms-23-08821-f008:**
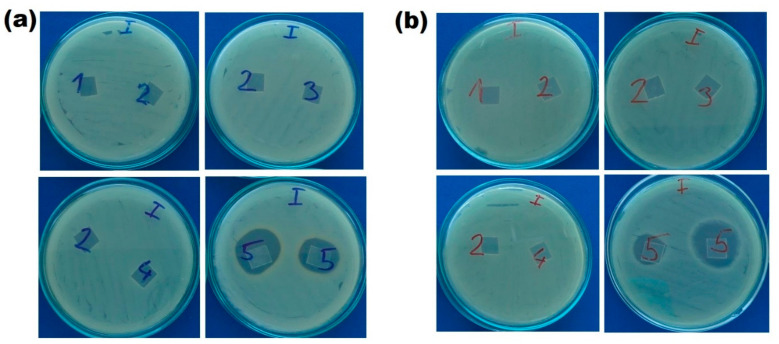
Zone of inhibitions around (1) untreated PLA (2) CS (3) *g*-CS (4) *g*-CS-ChS (5) lomefloxacin drug loaded *g*-CS-ChS coated PLA films against (**a**) *S. aureus* and (**b**) *E. coli*.

**Figure 9 ijms-23-08821-f009:**
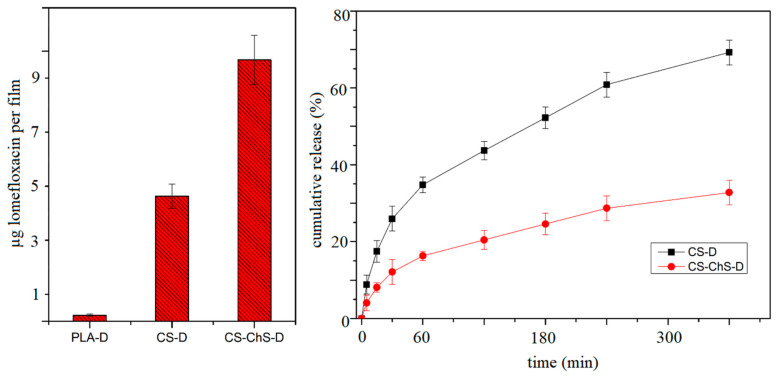
Drug loading and release profiles of CS and CS-ChS coated films.

**Table 1 ijms-23-08821-t001:** Values for water contact angle for the prepared films.

Sample Type	Contact Angle (%)
PLA	81.5 ± 1.4
RF	46.1 ± 0.8
CS	62.6 ± 1.2
*g*-CS	69.4 ± 1.6
*g*-CS-ChS	50.9 ± 0.9

**Table 2 ijms-23-08821-t002:** The atomic weight percentage of unmodified and modified PLA samples.

Sample Type	Composition (%)					Ratio
	C	O	N	F	S	N/C
PLA	70.9	28.4	-	-	-	-
RF	65.3	31.9	2.7	-	-	0.041
CS	77.6	19.4	3.0	-	-	0.044
*g*-CS	74.1	22.2	3.6	-	-	0.048
*g*-CS-ChS	60.9	26.6	10.1	-	0.5	0.165
*g*-CS-ChS-L	64.6	27.8	6.9	0.6	0.4	0.107

## Data Availability

The data presented in this study are available upon reasonable request from the corresponding author.

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
