# Peer review of "Effect of Saccharides Coating on Antibacterial Potential and Drug Loading and Releasing Capability of Plasma Treated Polylactic Acid Films"

_ijms, 2022, doi:10.3390/ijms23158821_

Round 1

Reviewer 1 Report

The paper is very well written. Clear guiding to the subject of interest, clear experimental section,

clear discussion.

Some proposals for very small improvements:

  • Please, check ml and mL, for example in lines 227, 228, 230, 231.
  • In line 232, I would prefer to set a blank between 24 and h.
  • In line 234, begin with no indent.
  • In Figs. 7, I would prefer larger digits. And no more than 5 numbers per axis. Please, write

small letters, not Time (min) but time (min)—with a blank.

Conclusion

I have never written a review that length. Well done!

Author Response

 Response to comments from Reviewer 1  

The paper is very well written. Clear guiding to the subject of interest, clear experimental section,

clear discussion.

Some proposals for very small improvements:

  • Please, check ml and mL, for example in lines 227, 228, 230, 231.
  • In line 232, I would prefer to set a blank between 24 and h.
  • In line 234, begin with no indent.
  • In Figs. 7, I would prefer larger digits. And no more than 5 numbers per axis. Please, write

small letters, not Time (min) but time (min)—with a blank.

Conclusion

I have never written a review that length. Well done!

We very much appreciate these helpful comments. As suggested by the reviewer, necessary changes about indents, blank between 24 and h, mL were done and all the paper was proofread to eliminate all such errors. We have changed Figures 7 (now Figure 9) as suggested.

Reviewer 2 Report

This manuscript investigated the “Effect of Saccharides Coating on Antibacterial Potential and Drug Loading and Releasing Capability of Plasma Treated Polylactic Acid Films. The subject of this paper is interesting and the results seem to support the enhancement of antibacterial activities. However, the data needs some clarification before acceptance. Thus, this manuscript needs major revision before acceptance. The list of commends is shown below.

1.     In XPS, the author should also provide a details scan of C1s and N1s to provide detailed chemical binding mechanisms and discuss.

2.     In XPS, Fig. 3C shows Ca2p and explains it. Is the sample containing impurities?

3.     “additive effect of both ChS and CS interaction” Could you explain this interaction in more detail for a clear understanding of antibacterial effects.

4.     Why S. aureus have good antibacterial activity than E. coli and its detailed discussion is missing?

5.     In Fig. 6 (a,b), sample number 5 shows different zone inhibition? Why same sample have different zone values? Is the drug loading/coating not stable or repeatable? The author may also provide the coating stability of the drug or polymer on the polymeric substrate.

6.     Moreover, the lomeflaxin-loaded sample only shows zone inhibition than other samples. Then what is the role of g-CS or g-CS-ChS? Is it possible to load the lomeflaxin directly to the polymeric substrate through coupling chemistry or physical loading?

7.     Author should provide SEM images after bacteria analysis of the film.

8.     Bacterial killing mechanism is not clear in this work.

9.     Extensive editing of English language and style required. Thereby, the language should be smoothed so that this article can be more readable and understandable.

Author Response

Response to comments from Reviewer 2  

This manuscript investigated the “Effect of Saccharides Coating on Antibacterial Potential and Drug Loading and Releasing Capability of Plasma Treated Polylactic Acid Films”. The subject of this paper is interesting and the results seem to support the enhancement of antibacterial activities. However, the data needs some clarification before acceptance. Thus, this manuscript needs major revision before acceptance. The list of commends is shown below.

  1. In XPS, the author should also provide a details scan of C1s and N1s to provide detailed chemical binding mechanisms and discuss.

C1s and N1S deconvolutions of the films provided as Fig. 4 and Fig. 5, and discussed in lines 320-347.

  1. In XPS, Fig. 3C shows Ca2p and explains it. Is the sample containing impurities?

As pointed by the reviewer, 3 samples contain Ca as impurities and the explanation added to the ‘Results and Discussion’ (lines 315-317) section.

  1. “additive effect of both ChS and CS interaction” Could you explain this interaction in more detail for a clear understanding of antibacterial effects.

Thank you for this suggestion. As suggested by the reviewer, the detailed explanation about ‘additive effect’ added to the ‘Results and Discussion’ (lines 426-448) sections.

  1. Why S. aureus have good antibacterial activity than E. coli and its detailed discussion is missing?

Thank you for this comment. As suggested by the reviewer, the detailed explanation about ‘additive effect’ added to the ‘Results and Discussion’ (lines 452-464) sections.

  1. In Fig. 6 (a,b), sample number 5 shows different zone inhibition? Why same sample have different zone values? Is the drug loading/coating not stable or repeatable? The author may also provide the coating stability of the drug or polymer on the polymeric substrate.

The different zone of inhibitions of the samples no.5 could be due to the inhomogeneous drug loading during complex formation between CS and ChS. We are going to include long-term stability and drug releasing tests in the following research. Thank you for your suggestion and we have included your point in the ‘Results and Discussion’ (lines 484-486) section.

  1. Moreover, the lomeflaxin-loaded sample only shows zone inhibition than other samples. Then what is the role of g-CS or g-CS-ChS? Is it possible to load the lomeflaxin directly to the polymeric substrate through coupling chemistry or physical loading?

Direct drug loading onto plasma treated films had been conducted and demonstrated on Fig.9. Drug loading efficiency of PLA without a proper substrate was too low. Apart from this, one of aims of the study was to show the potential of polysaccharides as antibacterial agents and evaluate their efficiency without antibiotic usage. This lack of clarity addressed in the sections of ‘Introduction’ (lines 66-70), ‘Antibacterial Efficiency of PLA Films’ (lines 477-478), ‘Inhibition Zone Assay’ (lines 496-498), ‘Drug Release’ (lines 508-520). In addition, the importance of CS in reversing the antibiotic resistant bacteria into re-sensitive state stated in the lines 115-118.

  1. Author should provide SEM images after bacteria analysis of the film.

For the detailed analysis on bacteria and to test the efficiency of our polysaccharide coated films on biofilm formations we are collaborating with Universitat de Barcelona as a following study.

  1. Bacterial killing mechanism is not clear in this work.

Bacteria killing mechanisms explained in detail in ‘Antibacterial Efficiency of PLA Films’ (lines 426-464) section.

  1. Extensive editing of English language and style required. Thereby, the language should be smoothed so that this article can be more readable and understandable.

Thank you for this suggestion. Several issues have been fixed.